# Assessing the effects of agricultural intensification on natural habitats and biodiversity in Southern Amazonia

**Jan Göpel***[�], **Jan Schüngel**[‡], **Benjamin Stuch**[‡], **Rüdiger Schaldach**[é]

Center for Environmental Systems Research (CESR), University of Kassel, Kassel, Germany

é These authors contributed equally to this work.
‡ These authors also contributed equally to this work.
* jan.goepel@usf.uni-kassel.de

**Data Availability Statement:** The data is available at https://doi.org/10.6084/m9.figshare.12014880.v1.

## Abstract

The ongoing trend toward agricultural intensification in Southern Amazonia makes it essential to explore the future impacts of this development on the extent of natural habitats and biodiversity. This type of analysis requires information on future pathways of land-use and land-cover change (LULCC) under different socio-economic conditions and policy settings. For this purpose, the spatially explicit land-use change model LandSHIFT was applied to calculate a set of high-resolution land-use change scenarios for the Brazilian states Para and Mato Grosso. The period of the analysis were the years 2010–2030. The resulting land-use maps were combined with maps depicting vertebrate species diversity in order to examine the impact of natural habitat loss on species ranges as well as the overall LULCC-induced effect on vertebrate diversity as expressed by the Biodiversity Intactness Index (BII). The results of this study indicate a general decrease in biodiversity intactness in all investigated scenarios. However, agricultural intensification combined with diversified environmental protection policies show least impact of LULCC on vertebrate species richness and conservation of natural habitats compared to scenarios with low agricultural intensification or scenarios with less effective conservation policies.

## 1. Introduction

Human induced changes to the biosphere have caused severe losses of biodiversity [1, 2]. Important factors leading to a loss of natural habitats are land-use and land-cover changes (LULCC) in particular due to the expansion and intensification of agriculture [3].

Martinelli et al. [4] show that, the growth of Brazil's agricultural sector from the 1970s until the end of the first decade of this century was an important driver of massive deforestation. During that period. an area of 18.8% of the original Brazilian Amazon biome, that is distinguished as a biodiversity hotspot [5], has been cut down [6] and converted to a large extend to cropland and pasture posing a major threat to terrestrial biodiversity [7, 8]. Agriculture plays an important role in regard to Brazils GDP (6.1%) [9]. Even more important, a share of 39% of Brazils exported goods are agricultural commodities, and products [10]. This strong

**Funding:** This study has been conducted as part of the Carbiocial project (funding reference number 01LL0902A-01LL0902N) commissioned by the German Federal Ministry of Education and Research (https://www.bmbf.de/). J.G., J.S. received respective funding. The funders had no role in study design, data collection and analysis, decision to publish, or preparation of the manuscript.

**Competing interests:** The authors have declared that no competing interests exist.

contribution of the agricultural sector to Brazil's overall economic performance has had positive impacts on social prosperity in the country. According to the Vieira et al. [11], the income of 29 million people has been considerably increased, lifting them out of poverty. Despite these positive numbers, the global demand for agricultural products is projected to continuously rise over the coming decades [12] driven by global population growth and increasing per capita demand for food, fodder, energy crops, and timber [13, 14]. Moreover, changes in food consumption patterns likely further increase food demands per capita [9]. These developments will most likely lead to further expansion and intensification of agricultural area in Brazil at the expense of natural habitats and biodiversity [15].

Among scholars, there is a controversial discussion regarding the impact of agricultural intensification on habitat and biodiversity loss. Some authors argue that intensification is key for a further increase in productivity [16], whereby the future destruction of natural habitats can be avoided by slowing down the expansion of agricultural land e.g. [17–19]. In contrast, there is the argument that agricultural intensification might even foster area expansion due to the so-called "rebound effect" [20] or increasing competiveness of agriculture and, thus higher attainable revenues [21]. The latter argument may only be applicable to situations where commodities with high demand elasticity are involved [22].

Simulation models and scenarios are effective tools to explore current and future land-use changes and to enhance the scientific understanding of their dynamics and drivers. A number of studies examine land-use changes in the Amazon region by employing different models e.g. [23–30]. Moreover, several studies assess the impacts of land-use changes on biodiversity in the tropics e.g. [31–33]. However, Chaplin-Kramer et al. [34] focus on scenarios of total forest conversion without taking into account specific socioeconomic developments. Laurance et al. [15] highlight the general causes of a loss of tropical nature and their implications without identifying specific regions in which conservation efforts might be needed to increase. Others focus their work on a single specific species as an indicator for biodiversity [2, 32, 33]. This approach has its eligibility but is limited to the perspective of that single species while, for instance, Ritter et al. [35] advise against the use of one taxonomic group as an indicator for species richness across different taxonomic groups or, in other words, biodiversity [36]. Newbold et al. [2] adopt a rather broad perspective in terms of methodology as well as spatial extent, thus they cannot identify specific drivers of biodiversity reduction or offer specific recommendations on how to confront possible implications. Widely used indicators for quantifying the effect of land-use change on biodiversity are the Mean Species Abundance (MSA) [37] and the Biodiversity Intactness Index (BII) [38]. Both indicators determine the impacts of land-use and its intensity level on species diversity and species abundance (as important aspects of biodiversity) and therefore are suitable tools for capturing the effects of agricultural intensification. Originally, the MSA was developed for global-scale analyses while the first application of BII was for a large-scale case study in Southern Africa (e.g. [39]). More recent articles propose the BII as a suitable indicator to measure the loss of species diversity in large-scale assessments [34, 40], in particular related to the Planetary Boundary concept [41].

The BII provides information to what extent vertebrate species abundance associated with each single grid-cell (900m x 900m) is influenced by LULCC by relating the average abundance of groups of organisms in current times (influenced by LULCC) to their average abundance in pre-industrial times [38]. This has the advantage of giving a broad overview of the effect of LULCC on vertebrate diversity on large to global scales. Lamb et al. [42] suggest that especially species abundance based indices perform very well as indicators for biodiversity change [42]. However, this broad view goes hand in hand with some downsides. One is the assumed (expert-based) and relatively coarse depiction of the degradation of natural habitats

expressed by the population impact, a factor that summarizes the total impact of the anthropogenic interference in natural ecosystems [43].

Main objective of the paper is to assess the potential effects of future agricultural intensification on habitat and biodiversity loss in the federal states Mato Grosso (MT) and Pará (PA) in Brazil. For this purpose, we analyze a set of spatially explicit LULCC scenarios with a time horizon until 2030, characterized by different assumptions regarding agricultural and socio-economic development in the region as well as by different environmental policies. The first part of our study determines the effects of the loss of natural habitat due to the expansion of agricultural area on the distribution ranges of vertebrate species. In the second part of our analysis we use the BII as an indicator to evaluate the combined effects of expansion and intensification of agriculture.

## 2. Material and methods

### 2.1. Study area

This study focusses on the two Brazilian federal states MT and PA (see Fig 1). These states differ greatly in respect to their recent agricultural developments and their level of exploitation of natural habitats due to the Brazilian agricultural development frontier running through this region [44, 45].

PA has an area of 1.25 million km$^2$ and a population of 8 million people [46]. Only 11,969 km$^2$ of the land is used for soybean cultivation [46]. In 2015 1,881 km$^2$ were deforested which is about the same amount of deforestation as in 2014 [6]. The dominant land use sector is cattle ranching with a total herd size of 19.2 million animals [46]. A hot spot of LULCC is along the Cuiabá-Santarem highway (BR-163), the most recent of the "development highways" which are used to acquire the agriculturally rather underdeveloped northern parts of Brazil for crop cultivation and cattle ranching [47]. The natural vegetation is dominated by dense rainforest [11] covering about 77.6% of the state's area according to MODIS land cover data [48]. More than 40,000 vascular plant species can be found here, of which 30,000 are endemic [11]. Over 1,000 bird species are harbored in the Amazon biome [11] as well as a high concentration of mammals, of which many are endemic, especially along the courses of the rivers crossing this biome [5]. Of the 875 amphibian species in the country, approximately 50% are concentrated in the Amazon biome [5]. Especially here the potential for a loss of vertebrate diversity is high due to a high density of endemic, threatened, and small ranged species [5] as well as ongoing and expected future agricultural expansion [15].

MT has an area of 907,000 km$^2$ and a population of 3.2 million inhabitants [46]. 69,807 km$^2$ of land is used for soybean cultivation [46] and 1,508 km$^2$ were deforested in 2015 which constitutes an increase of 16% in comparison to 2014 [6]. Another dominant land use sector is cattle ranching with a total herd size of 28.4 million animals [46]. Here the expansion of area used for soybean cultivation and cattle ranching could be identified as the primary cause of conversion of natural ecosystems to agricultural land [49, 50]. In comparison to PA, MT is more consolidated in terms of agricultural expansion. In recent years, the steep decline of availability of highly productive farmland, policies to curb deforestation and rising land prices have led to a development toward agricultural intensification and away from agricultural expansion [18, 51, 52]. MT is covered by two Brazilian biomes, the Brazilian Cerrado and the Amazon rainforest [5, 45]. Here, 7,000 plant species, of which 44% are endemic to the Cerrado, can be found. The Cerrado biome is especially rich in bird diversity with 837 species, which resembles 49% of all bird species found in Brazil. Also 150 reptile species (50% of all Brazilian reptile species) and 180 amphibian species (28% endemic to the Cerrado) are found here. The Cerrado biome and its waterways are home to 1,400 fish species, 40% of all fish species occurring in Brazil [53].

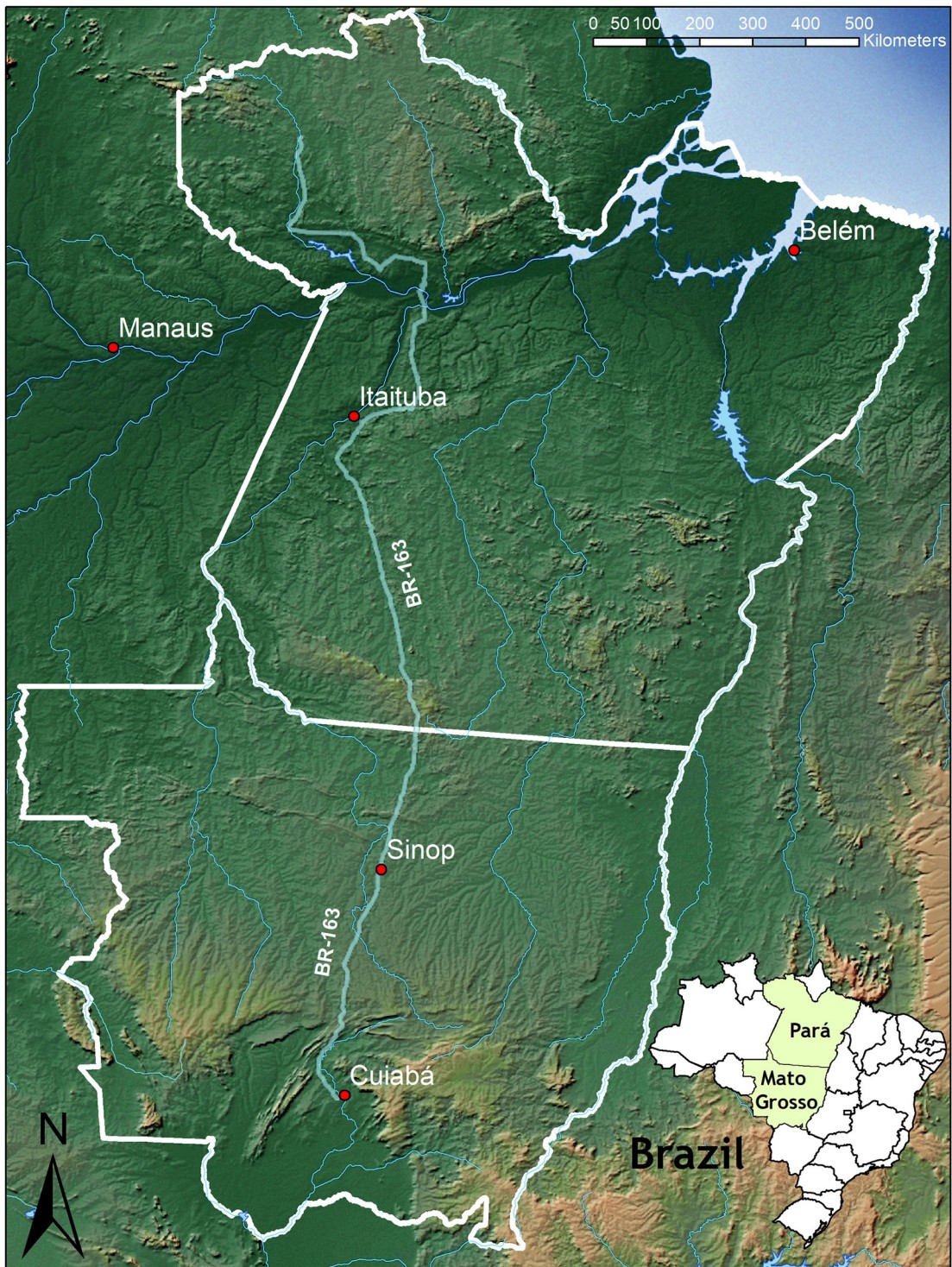

**Fig 1. Overview map of the study region (Pará and Mato Grosso in Brazil).**

## 2.2. Modeling and assessment protocol

LULCC scenarios (section 2.3) were generated with the spatially explicit LandSHIFT model. The model is fully described in [54] and has been tested and validated in different case studies for Brazil [26, 55, 56]. It is based on the concept of land-use systems [57] and couples components that represent the respective anthropogenic and environmental sub-systems. In our study, LULCC was simulated on a raster with the spatial resolution of 900m x 900m that covers the territories of the federal states of MT and PA. A Table elaborately describing the input data used to initialize and run the model can be found in [58].

The LandSHIFT model generates digital maps for 2010 until 2030 in 5-year time steps that depict the resulting LULCC. For further analysis, we aggregated the land-use types used by the model (Table 1): The 12 crop types [54] were aggregated into the land-use class cropland, the 5 forest types [48] into the class rainforest, and the 2 savannah types [48] into the class Cerrado. Changes in location and area of the respective land-use types were determined by comparing the maps for 2010 and 2030 using GIS software.

As a second step, we merged the simulated land-use maps of each calculated scenario with maps of vertebrate species distribution ranges [5] regarding three taxa and three categories by overlaying the land-use and land-cover maps with spatially explicit maps of vertebrate species distribution ranges, again using GIS software. Through this, we could correlate the species distribution ranges to natural habitats. Further, this enables quantifying the impact of simulated LULCC on natural habitat area (conversion of natural habitats) known to domicile vertebrate species.

Finally, we calculated the Biodiversity Intactness Index (BII) for the reference year 2010 and 2030 according to eq 1. This was accomplished by assigning each land-use type a specific population impact and multiplying this population impact by the number of grid cells covered by the respective land-use type and the vertebrate species abundance (per taxon and category) associated with these grid cells.

## 2.3. Land-use scenarios

In order to explore agricultural intensification and expansion in respect to different socio-economic and policy assumptions, 4 scenarios have been employed for modeling land use change. These scenarios have been developed during an interdisciplinary research project (CarBioCial; www.carbiocial.de) thematically covering the study area (MT, PA). They describe plausible future development pathways of Southern Amazonia until the year 2030. Each scenario consists of a qualitative part (storyline) that provides a short narrative of the future world and a set of quantitative information that describe the respective main drivers of LULCC [59, 60]. The storylines are elaborated by [61].

**Table 1. Aggregation of LandSHIFT land-use types.**

| LandSHIFT land-use types | aggregated land-use types |
|---|---|
| evergreen needle forest, evergreen broad-leafed forest, deciduous needle forest, deciduous broad-leafed forest, mixed forest | rainforest |
| closed shrub land, open shrub land | shrub land |
| woody savannah, savannah | savannah (Cerrado) |
| tea, cocoa, coffee, maize, annual oil crops, pulses, rice, tropical roots and tubers, soybean, sugarcane, cassava, wheat | cropland |
| rangeland, pasture | pasture |

The following paragraphs briefly describe the central assumptions of the scenarios. For a comprehensive overview of the quantitative scenario assumptions (crop production, crop yield, population, and livestock) see [56]. An elaborate description and discussion of the translation process from qualitative to quantitative information and the respective results for all scenario assumptions is described in Schönenberg et al. [62].

The *Trend scenario* describes a growing demand for agrarian products based on an extrapolation of growth trends from 1973 to 2000 specific for each modelled crop. Furthermore, it is assumed that environmental policies like the Brazilian Forest Code or the Soy- and Cattle Moratorium will not be implemented. Only the illegal conversion of natural habitats (protected areas) is prohibited due to good law enforcement. The technological development of agricultural practices in the study area includes an intensification of agricultural production through increasing crop yields. The possibility to intensify pasture management is not considered in this scenario.

Two intensification scenarios (Legal Intensification and Illegal Intensification) assume a growing demand for agrarian products (see *Trend Scenario*) further reinforced by population and GDP growth generated in Asian countries (as export markets). The technological developments of agricultural practices in the study area include a high degree of agricultural intensification including the intensification of pasture management. Both scenarios presume the intensification of cattle ranching. In PA, we assume an intensification rate of 4.5% per time step up to a maximum of 30%. That means that the biomass productivity of any pasture grid cell is increased by 4.5% until biomass productivity is 30% higher than in the base year. As agriculture in MT is presumed to be more mechanized, large scale, and world market oriented [63, 64], we assume an intensification rate of 9% up to a maximum value of 50%. These assumptions are based on observed pasture intensification rates in Brazil. According to Wint et al. [65] and Lapola et al. [17], the stocking density of pastures in Brazil rose continuously from 1990 to 2010, with a total increase of 45% during that period. The two scenarios vary in terms of environmental law enforcement. While the *Legal Intensification scenario* assumes compliance with environmental policies (environmental protected areas, *Brazilian Forest Code*), the *Illegal Intensification scenario* presumes noncompliance with environmental law expressed as the defiance of environmental protected areas concerning agricultural expansion and the noncompliance with the *Brazilian Forest Code*. This scenario assumes the possibility to convert land that is under conservation (e.g. nature reserves), thus opening up spaces that are not allowed for conversion in all other scenarios.

The *Sustainable Development scenario* describes a new social model. This new model includes citizenship, an inclusive economic system, clear land tenure rights, and strong law enforcement including participatory monitoring of deforestation. Furthermore, it portrays a substantial change in terms of anthropogenic consumption pattern, away from a meat oriented diet toward a healthy and sustainable diet as proposed by the WHO [66, 67] including further intensification of crop production. Moreover, the conversion of areas classified as covered by rainforest into agricultural area is not allowed according to the assumptions of the scenario.

## 2.4. Maps of vertebrate diversity and Biodiversity Intactness Index (BII)

We use maps of vertebrate diversity covering the whole area of Brazil [5] to illustrate the overlapping of areas of vertebrate diversity and simulated LULCC in each investigated scenario. The species diversity maps were generated by deriving polygon range data concerning birds from BirdLife International and NatureServe [68] and polygon range data concerning mammals and amphibians from the International Union for the Conservation of Nature [69]. These polygon range datasets were rendered at a spatial resolution of 10×10 km in order to

produce species diversity maps considering these three groups of terrestrial vertebrates in Brazil [5]. These groups were further subdivided into the categories small-ranged species, threatened species, and endemic species. Small-ranged species were defined as those species that have a range smaller than the median for that taxon (2,250,813 km$^2$ for birds, 1,230,901 km$^2$ for mammals, 66,979 km$^2$ for amphibians) in Brazil. For example, a bird species is considered to be small-ranged by occurring naturally in a range of less than 2,250,813 km$^2$, which resembles the median distribution range for that taxon in Brazil. Threatened species were defined as vulnerable, endangered, or critically endangered according to the IUCN Red List [70]. Considered threatened species are globally threatened as the use of taxanomies was inconsistent in the national RedList [5]. Endemic species were defined as having at least 90% of their range within Brazil and no part of their range extending more than 50 km beyond the Brazilian border. Overall, 1703 bird species, 637 mammal species and 875 amphibian species were considered in this study. We decided against the inclusion of the category total species richness in our assessment. Total species richness as an indicator for biodiversity can be misleading as it is mainly driven by wide-ranged species [5, 71] which might even be benefit from degraded habitats [72] while especially endemic and small-ranged species are dependent on the intactness of their respective ecosystems [73].

We calculated the BII accroding to [38] in order to assess the impact of LULCC on overall vertebrate diversity in the time from 2010 to 2030. The BII is defined as the population of a species group *i* under land-use activity *k* in ecosystem *j*, relative to a reference population on the same ecosystem type according to eq 1.

$$BII = \frac{\Sigma_i \Sigma_j \Sigma_k R_{ij} A_{jk} I_{ijk}}{\Sigma_i \Sigma_j \Sigma_k R_{ij} A_{jk}} \qquad (1)$$

Eq 1: Biodiversity Intactness Index [38].

$I_{ijk}$, the *"population impact"*, is the population of a certain species group *i* under land-use activity *k* in ecosystem *j*. $A_{jk}$ is the area of land-use *k* in ecosystem *j*, $R_{ij}$ the number of species of taxon *i* in ecosystem *j*.

Since the calculation is done on grid-cell level, each cell is associated with a specific land use type. The number of species is the sum of bird species, mammal species, and amphibian species assigned to one cell respectively. In order to formulate the population impact, a combination of impact values from [74, 75] and [39] was employed. These values indicate the reduction of mean species diversity in respect to a certain type of land use. The values employed are shown in Table 2. A BII value of 1 indicates a species abundance on the pre-colonial level. An index of 0,5 indicates that the species abundance is reduced by half in reference to the pre-colonial level.

A decreasing BII value is an expression for further reduction of biodiversity intactness due to LULCC affecting regions characterized by the occurrence of different species of different taxa. An increasing BII value expresses a recovery of biodiversity intactness mainly due to the displacement of anthropogenic land-use out of these regions or by replacement of certain land-use types by "less harmful" land-use types (e.g. cropland to fallow land) within these regions.

## 3. Results & discussion

The first section summarizes the main LULCC characteristics for each scenario. Thereafter, we assess the resultant effect on vertebrate species diversity, (1) by relating natural habitat area conversion and vertebrate species diversity as well as distribution ranges and (2) by calculating the BII.

**Table 2. Values used as population impact to calculate BII.**

| land-use | impact | source |
|---|---|---|
| Cropland | 0.15 | [74] weighted by proportion of high input agriculture to low input agriculture in Latin America |
| Pasture extensive (PA) | 0.6 | [75] |
| intensive (MT) | 0.3 | |
| Mosaic Agricultural Area/rainforest (Legal Reserve in the transition matrix) | 0.83* | 20% cropland impact as calculated above and 80% undisturbed forest impact [74] |
| Rainforest | 1.0 | [74] |
| Grassland, Savannah, Shrubland, Wetland (natural vegetation in the transition matrix) | 0.94 | [39, 76] |
| Fallow land | 0.5 | [74] |
| Urban | 0.05 | [74] |

* A population impact value of 0.83 has been assumed for areas in the Amazon biome that are made up of 20% cropland and 80% rainforest, the so called "Legal Reserve", in which any kind of deforestation is prohibited according to the *Brazilian Forest Code* [77, 78]. This population impact value is considered as an expression for the fragmentation of rainforest.

## 3.1. Area conversion in the land-use change scenarios

In the following, the main LULCC characteristics of each scenario are described in form of a land-change matrix (Table 3).

The simulation of the *Trend Scenario* led to a strong conversion of rainforest to pasture area in PA while a smaller share of rainforest was converted into cropland. Only relatively small shares of other natural vegetation had to be converted into managed land. In MT, a comparatively large area covered by natural vegetation had to be converted into pastures and a smaller area into cropland. Rainforest was only converted into pastures. An intensification of pasture management is not part of this scenario.

In contrast, the *Legal Intensification Scenario* led to no conversion of rainforest into pastures in both Federal States. This is explained by the intensification of pasture management as described in section 2.3. Nonetheless, in PA the conversion of rainforest into cropland was increased by 221% in comparison with the *Trend Scenario* due to an increasing demand for, on the one hand, fodder crops (soybean) and, on the other hand, staple crops used for human consumption in Asian countries [61]. In addition to the rainforest that is converted to cropland, 216,020 km$^2$ of rainforest are classified as Legal Reserve in PA. This land is not converted from its original vegetation cover but is considered disturbed due to being fragmented by small-scale agriculture. Thus, it is considered with a population impact of 0.83 (see Table 3) when calculating the BII for PA. In MT, no rainforest was converted into managed land while the conversion of natural vegetation into managed land could be decreased by 3.1% compared to the *Trend Scenario*.

In the case of the *Illegal Intensification Scenario*, in PA the conversion of rainforest to cropland could be reduced by 99% and the conversion of rainforest to pastures could be decreased by 26% in comparison with the *Trend Scenario*. In PA, particularly pastures are relocated into protected areas covered by rainforest due to higher net primary productivity. In MT, a strong relocation of cropland into protected areas is discernible. Here, possible crop yields are higher than in other regions of the Federal State.

In the case of the Sustainable Development Scenario, the conversion of rainforest into managed land was not allowed (see section 2.3). The shift of the consumption behavior towards a

**Table 3. Land-use and land-cover change matrix for the years 2010 to 2030 ($10^3$ km$^2$).**

| PA | TREND | | | LI | | | | ILI | | | SUST | | |
|---|---|---|---|---|---|---|---|---|---|---|---|---|---|
| | CR | PS ext. | UR | CR | PS ext. | UR | LR | CR | PS ext. | UR | CR | PS ext. | UR |
| RF | 16.84 | 89.94 | 0.00 | 54.01 | 0.00 | 0.00 | 216.02 | 33.48 | 66.15 | 0.00 | 0.00 | 0.00 | 0.00 |
| NV | 5.82 | 6.38 | 0.01 | 8.95 | 5.00 | 0.01 | 0.00 | 8.84 | 6.29 | 0.01 | 4.80 | 0.00 | 0.01 |
| FA | 0.19 | 0.00 | 0.00 | 0.34 | 0.00 | 0.00 | 0.00 | 0.04 | 0.40 | 0.00 | 0.48 | 0.00 | 0.00 |
| CR | 139.33 | 0.00 | 0.03 | 139.33 | 0.00 | 0.03 | 0.00 | 128.62 | 4.47 | 0.03 | 125.26 | 0.00 | 0.02 |
| PS | 0.00 | 96.70 | 0.00 | 7.11 | 89.60 | 0.00 | 0.00 | 5.28 | 91.38 | 0.00 | 83.86 | 12.84 | 0.00 |
| UR | 0.00 | 0.00 | 0.56 | 0.00 | 0.00 | 0.56 | 0.00 | 0.00 | 0.00 | 0.56 | 0.00 | 0.00 | 0.56 |
| MT | TREND | | | LI | | | | ILI | | | SUST | | |
| | CR | PS int. | UR | CR | PS int. | UR | LR | CR | PS int. | UR | CR | PS int. | UR |
| RF | 0.00 | 28.38 | 0.00 | 0.00 | 0.00 | 0.00 | 0.00 | 0.70 | 0.00 | 0.00 | 0.00 | 0.00 | 0.00 |
| NV | 6.62 | 27.82 | 0.00 | 18.51 | 17.05 | 0.00 | 0.00 | 51.26 | 3.33 | 0.00 | 40.10 | 0.00 | 0.00 |
| FA | 0.00 | 0.00 | 0.00 | 0.00 | 0.00 | 0.00 | 0.00 | 0.00 | 0.00 | 0.00 | 0.07 | 0.00 | 0.00 |
| CR | 184.63 | 23.47 | 0.00 | 208.52 | 0.00 | 0.00 | 0.00 | 181.51 | 5.90 | 0.00 | 208.51 | 0.00 | 0.00 |
| PS | 0.00 | 158.42 | 0.00 | 3.05 | 148.13 | 0.00 | 0.00 | 1.36 | 132.62 | 0.00 | 2.31 | 156.10 | 0.00 |
| UR | 0.00 | 0.00 | 0.85 | 0.00 | 0.00 | 0.85 | 0.00 | 0.00 | 0.00 | 0.85 | 0.00 | 0.00 | 0.85 |

(TREND = Trend Scenario, LI = Legal Intensification Scenario, ILI = Illegal Intensification Scenario, SUST = Sustainable Development Scenario) (CR = cropland, PS (extensive (ext.) in PA; intensive (int.) in MT) UR = urban area, RF = Rainforest, NV = natural vegetation, FA = fallow land).

diet that is mainly based on crop products helped to avoid the necessity to convert rainforest or other natural vegetation into pastures. The reduction of meat demand led to a release of pasture area that could be utilized to expand cropland area that is now employed to produce additional crops in order to substitute the share of calories that was originally consumed in the form of meat in PA. In order to realize this substitution of calorie sources in MT, compared with the *Trend Scenario*, 16% more natural vegetation had to be converted into cropland.

Agricultural intensification has played an important role in regard to recent agricultural production growth in Brazil and is likely to further increase Brazil´s crop and beef production considerably [79]. The observed decoupling of production increases from deforestation in the latter half of the first decade of this century [29, 80] have shown that the intensification of agricultural systems not only supports food provisioning, it also limits the expansion of agricultural area; thus the destruction of natural habitats [81]. This trend could be confirmed. The negative effect of projected future agricultural production growth on the extent of natural habitats is considerably reduced through agricultural intensification and particularly through intensification of grazing intensities on pastures (compare with [19, 81]). Especially the Legal Intensification Scenario shows substantially less converted natural habitats compared to the *Trend Scenario* based on constant crop yields and grazing intensities of the reference year 2010. This is also confirmed by Cohn et al. [18] who have shown that the encouragement of an intensification of pastures, either through subsidies of intensified systems or tax on extensive pastures, considerably limits the conversion of natural habitats until 2020. Nonetheless, the reducing effect of an intensification of pastures is much lower in the case of the *Illegal Intensification Scenario* where protected areas are made accessible for conversion into managed land. This leads to a relocation of managed land into these areas due to higher net primary productivity on pastures or higher possible yields on cropland, causing higher conversion rates. This illustrates what has been confirmed by several other authors e.g. [82, 83], who found that the

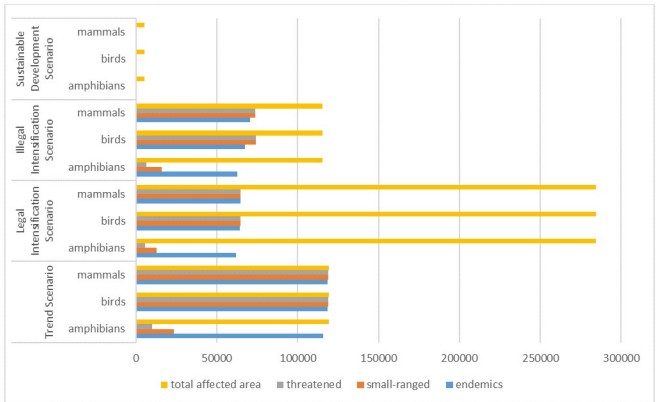

**Fig 2. Natural habitat area of investigated vertebrate species affected by conversion over total converted area in Pará.**

intensification of agricultural production, if it goes hand in hand with the protection of natural habitats, have the highest impact in terms of limiting the conversion of natural habitats, and thus, promoting the conservation of vertebrate species diversity.

## 3.2. Effects of natural habitat loss on the distribution ranges of vertebrate species

**3.2.1. Pará.** Fig 2 illustrates that the highest effect in terms of a distrubance of natural habitats and distribution ranges of vertebrate species in PA can be expected in case of the *Trend Scenario*.

Allmost all of the 119,185 km$^2$ of converted natural habitats in the case of the *Trend Scenario* is identified as a distribution range of bird and mammal species. Also, close to all of this coverted area is a ditribution range of endemic amphibians. Only the distribution ranges of small-ranged and threatened aphibians are not affected strongly by the conversion of natural habitats into managed land.

67,914 km$^2$ of natural habitats were converted in the case of the *Legal Intensification Scenario* excluding 216,022 km$^2$ of Legal Reserve as it remains in its original land-cover type "Rainforest". The calculation of the area-weighted change of affected distribution ranges, including the Legal Reserve as it is considered affected by fragmentation, could be reduced by 46% for all mammal and bird categories. The afflicted distribution ranges could be decreased by 47% for endemic amphibians, by 45% for small-ranged and by 44% for threatened amphibians compared to the situation in the *Trend Scenario*.

In the case of the *Illegal Intensification Scenario*, 114,608 km$^2$ of natural habitat area is converted into managed land. In this case, the calculation of the area-weighted change shows a reduction of the affected distribution ranges by 40% for endemic bird and mammal species and by 38% for all other bird and mammal categories in comparison to the *Trend Scenario*. Also, the afflicted distribution ranges of endemic, small-ranged and threatened aphibians could be decreased by 46%, 31% and 37% respectively.

The least negative effect on vertebrate species diversity due to a conversion of natural habitats is discernible in the case of the *Sustainable Development Scenario*. Not only the area of converted natrual habitats is less as compared to the other scenarios, also the share of this area that is a known habitat to vertebrate species is relatively low. The affected area known as

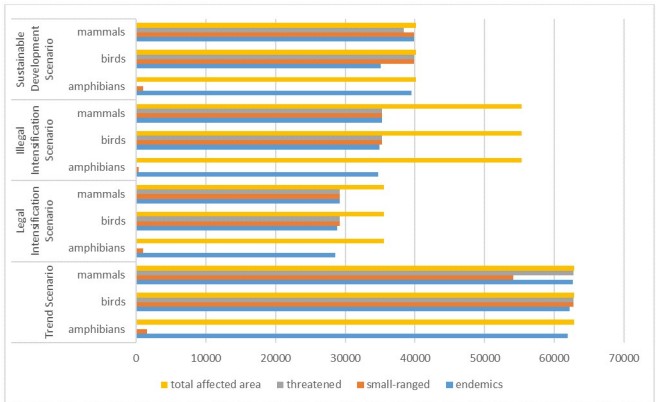

**Fig 3. Natural habitat area of investigated vertebrate species affected by conversion over total converted area in Mato Grosso.**

distribution range of all investigated vertebrate species could be reduced by over 99% (area-weighted change) compared to the *Trend Scenario*.

**3.2.2. Mato Grosso.** As Fig 3 shows, the assumptions made for the *Trend Scenario* in MT result in a strong disturbance of vertebrate species diversity.

An area of 62,824 km$^2$ of natural habitats is converted in the case of the *Trend Scenario*. An especially strong impact can be seen for all categories of bird species, endemic and threatened mammals and endemic amphibians. The distribution range of small-ranged mammals is slightly less affected. Small-ranged amphibian distribution ranges are least afflicted. The situation concerning threatened amphibians (for all scenarios) may not be consistent with the actual situation as threatened amphibians are especially data deficient in Brazil [5].

In the case of the *Legal Intensification Scenario* 35,555 km$^2$ of natural habitats were converted into managed land. The calculation of the area-weighted change of affected distribution ranges illustrates a reduction of 53% for all amphibian, bird and mammal taxa and categories except small ranged amphibians (-30%) and small-ranged mammals (-46%) in comparison with the *Trend Scenario*.

Concerning the *Illegal Intensification Scenario*, the negative effect of a conversion of natural habitats (55,301 km$^2$) on the distribution ranges of all investigated taxa and categories per taxon could be reduced (area-weighted change) by 44%, except small-ranged amphibians (-73%) and small-ranged mammals (-35%) despite the fact that the total converted area was only reduced by 12% in comparison with the *Trend Scenario*.

The assumptions made in the case of the *Sustainable Development Scenario* lead to a reduction of converted natural habitats to 40,166 km$^2$. The positive effect expressed as a reduction of the afflicted distribution ranges (area-weighted change) amounts to 36% for endemic amphibians and mammals as well as small-ranged and threatened birds. This positive effect can also be witnessed for small-ranged amphibians (-32%), endemic birds (-44%), small-ranged (-26%) as well as threatened mammals (-38%).

**3.2.3. Factors influencing natural habitat loss and the distribution ranges of vertebrate species: A scenario comparison.** In MT, the area of affected natural habitats in the case of the *Illegal Intensification scenario* could be reduced by 12% compared to the *Trend Scenario*. This reduction of transformed natural habitats due to agricultural intensification is surpassed in the *Legal Intensification Scenario*. Here, the conversion of natural habitats could be

decreased by 43% compared to the *Trend Scenario*. Also in both cases, the affected distribution ranges of vertebrate species could be reduced considerably. These results suggest that intensification measures are effective. This is confirmed by Strassburg and Latawiec who found in their study of possible future agricultural productivity scenarios that an increase of pasture productivity to 49–52% of its potential productivity in Brazil would suffice for the projected increase of agricultural production without further appropriation of natural habitats [84]. Although, the effect of combined intensification and adequate conservation policies can contribute stronger to the preservation of the assessed vertebrate species as the results of the *Legal Intensification Scenario* (see Section 2.3) show. This can be confirmed by Sparovek et al. who indicate that in all of their simulated future land use scenarios, effective conservation policies played the most important role in conserving natural habitats [85]. Further, the effective protection and conservation of forested land will not impair Brazil´s ability to produce agricultural ressources for demanding markets in the near future as has been confirmed by de Souza Ferreira Filho et al. [86]. This picture is confirmed in PA. Here, the amount of converted natural habitats could be limited by 43% (*Legal Intensification Scenario*) and by 3% in the case of the *Illegal Intensification Scenario*. Despite this relatively low reduction of converted natural habitats in the latter case, this simulated decrease still led to a considerable reduction of afflicted distribution ranges of vertebrate species due to a spatial shift of habitat conversions away from the distribution ranges of the investigated vertebrate species. Concerning the *Sustainable Development Scenario* in MT, cropland area expands especially strong due to a shift of anthropogenic consumption patterns away from meat toward crop intake while there is only a slight release of pasture area. Therefore, the decrease of pasture area can only partially counteract the expansion of cropland, and thus the loss of natural habitat area in MT remains relatively high compared to the situation in PA. This is also expressed in the reduction of affected distribution ranges. The simulated relocation of cropland onto released pasture area in PA almost completely prevents the loss of natural habitats that are known distribution ranges of the investigated vertebrate species. This development is confirmed by Alkimim et al. who found that 50 million hectares of pasture land in Brazil are suitable for crop prodcution and thus, can spare forest area from conversion into agricultural land [87]. Especially here, the shift of human consumption toward a plant based diet is the key factor for limiting the loss of natural habitats resembling distribution ranges of vertebrate species as the intensification of pastures is not part of the assumptions for the Sustainable Development Scenario. The optimal combination of intensification and conservation measures as well as the interplay with changes of anthropogenic consumption patterns with the goal of a maximized reduction of converted natural habitats depends on the present situation in the respective region. This is illustrated by the heterogeneity of conversions of natural habitats in PA and MT, under the respective scenario assumptions.

## 3.3. The combined impacts of expansion and intensification of agriculture on vertebrate diversity

**3.3.1. Pará.** The effects of a conversion of natural habitats on vertebrate species diversity are confirmed by our assessment of the BII in PA. Table 4 shows, the impact on species diversity, as expressed by changes of the BII, is strongest as calculated in the case of the *Trend Scenario* in PA.

We found especially strong redcutions of BII for small-ranged amphibians, threatened mammals, followed by threatened amphibians and endemic mammal species. In the case of the *Legal Intensification Scenario*, we see BII value decreases for all taxa and categories. Especially strong disturbances can be discerned in the case of small-ranged amphibian species,

**Table 4. Changes of BII in Pará between 2010 and 2030.**

| taxon | category | Trend 2010 | Trend 2030 | rel. Change [%] | LI 2030 | rel. Change [%] | ILI 2030 | rel. Change [%] | SD 2030 | rel. Change [%] |
|-------|----------|------------|------------|-----------------|---------|-----------------|----------|-----------------|---------|-----------------|
| Amphibians | endemic | 0.79 | 0.71 | -10.1 | 0.71 | -10.1 | 0.73 | -7.6 | 0.79 | 0 |
|  | small-ranged | 0.65 | 0.52 | -20 | 0.53 | -18.5 | 0.57 | -12.3 | 0.66 | 1.5 |
|  | threatened | 0.59 | 0.49 | -16.9 | 0.52 | -11.9 | 0.53 | -10.2 | 0.58 | -1.7 |
| Birds | endemic | 0.8 | 0.71 | -11.3 | 0.71 | -11.3 | 0.73 | -8.8 | 0.8 | 0 |
|  | small-ranged | 0.85 | 0.77 | -9.4 | 0.78 | -8.2 | 0.78 | -8.2 | 0.86 | 1.2 |
|  | threatened | 0.79 | 0.72 | -8.9 | 0.73 | -7.6 | 0.75 | -5.1 | 0.79 | 0 |
| Mammals | endemic | 0.79 | 0.68 | -13.9 | 0.69 | -12.7 | 0.71 | -10.1 | 0.79 | 0 |
|  | small-ranged | 0.86 | 0.77 | -10.5 | 0.78 | -9.3 | 0.79 | -8.1 | 0.86 | 0 |
|  | threatened | 0.79 | 0.65 | -17.7 | 0.69 | -12.7 | 0.72 | -8.9 | 0.78 | -1.3 |

(Trend = Trend Scenario, LI = Legal Intensification Scenario, ILI = Illegal Intensification Scenario, SD = Sustainable Development Scenario).

endemic mammals as well as threatened mammals. In the case of the *Illegal Intensifcation Scenario* especially small-ranged amphibian species, endemic mammals, and threatened amphibian species were strongly affected. The lowest negative effect was simulated in the case of the *Sustainable Development Scenario*. The highest decrease was calculated for threatened amphibians while the BII and threatened mammals decreased. All other BII values remained constant or even increased as is the case for small-ranged amphibians and small-ranged bird species. This slight increase can be attributed to the conversion of unused cropland cells to pasture cells.

**3.3.2. Mato Grosso.** Table 5 shows, the impact on species diversity in MT is more moderate in the case of the *Trend Scenario* compared to the situation in PA.

The highest reduction of BII in MT was simulated in the case of the *Trend Scenario*. Here, threatened bird species, threatened mammals, and endemic mammals are especially affected. In the case of the *Legal Intensification Scenrio*, we see decreasing BII values for all taxa and categories with the exception of small-ranged birds which remains constant. We found especially strong decreases for threatened mammals and threatened birds. Concerning the *Illegal Intensification Scenario*, especially small-ranged amphibians, threatened mammals and threatened bird species are impacted strongly. Interestingly, the *Sustainable Development Scenario* in MT results in a strong negative impact. It becomes obvious that all taxa and categories are affected

**Table 5. Changes of BII in Mato Grosso between 2010 and 2030.**

| taxon | category | Trend 2010 | Trend 2030 | rel. Change [%] | LI 2030 | rel. Change [%] | ILI 2030 | rel. Change [%] | SD 2030 | rel. Change [%] |
|-------|----------|------------|------------|-----------------|---------|-----------------|----------|-----------------|---------|-----------------|
| Amphibians | endemic | 0.67 | 0.62 | -7.6 | 0.66 | -1.5 | 0.67 | 0 | 0.62 | -7.5 |
|  | small-ranged | 0.56 | 0.54 | -3.6 | 0.55 | -1.8 | 0.36 | -35.7 | 0.54 | -3.6 |
|  | threatened | n.a. | n.a. | n.a. | n.a. | n.a. | n.a. | n.a. | n.a. | n.a. |
| Birds | endemic | 0.66 | 0.6 | -9.1 | 0.65 | -1.5 | 0.67 | 1.5 | 0.61 | -7.6 |
|  | small-ranged | 0.62 | 0.57 | -8.1 | 0.62 | 0 | 0.6 | -3.2 | 0.59 | -4.8 |
|  | threatened | 0.59 | 0.51 | -13.6 | 0.55 | -6.8 | 0.5 | -15.2 | 0.53 | -10.2 |
| Mammals | endemic | 0.63 | 0.57 | -9.5 | 0.62 | -1.6 | 0.64 | 1.6 | 0.59 | -6.4 |
|  | small-ranged | 0.66 | 0.6 | -9.1 | 0.65 | -1.5 | 0.64 | -3.0 | 0.63 | -4.6 |
|  | threatened | 0.68 | 0.61 | -10.3 | 0.63 | -7.4 | 0.6 | -11.8 | 0.63 | -7.3 |

(Trend = Trend Scenario, LI = Legal Intensification Scenario, ILI = Illegal Intensification Scenario, SD = Sustainable Development Scenario).

negatively, especially endemic amphibians and endemic birds as well as threatened birds and threatened mammals.

### 3.3.3. The combined effects of agricultural expansion and intensification on vertebrate diversity: A scenario comparison.

In general, when looking at sections 3.3.1. and 3.3.2. as well as the following discussion of the illustrated results, one has to keep in mind that the BII values in MT are on average 0.14 points below those calculated for PA as MT is more consolidated in agricultural terms [51, 52]. Newbold et al. [7] calculated BII values of around 70% for the Brazilian Cerrado (mainly located in MT) as well as 85% for the Amazon biome (mainly located in PA). This agrees well with our calculation for the year 2010 of 0.59–0.68 (MT) and 0.65–0.86 (PA) respectively (see Tables 4 and 5). The fact that the estimates found in our study are slightly lower than those estimated by Newbold et al. [7] is explained by taking into account that they focused on the whole Cerrado and Amzonia region. We assess a sub-region that is and was characterized by especially strong LULCC dynamics.

Interestingly in PA, the negative effect in the case of the *Illegal Intensification Scenario* is lower compared to the *Legal Intensification Scenario*. This can be explained by the consideration of the 216,022 km$^2$ of rainforest share of the total Legal Reserve area (270,027 km$^2$) in PA as affected area with a population impact of 0.83 (see Table 3). Here, the fragmentation of the total Legal Reserve area by small scale agriculture, and consequently the degradation of rainforest habitats e.g. [88, 89], has a negative impact that leads to a stronger decrease of the BII as compared to the *Illegal Intensification Scenario*. This is true despite the fact that in the *Illegal Intensification Scenario* a larger amount of natural habitat area is actualy converted into managed land with a higher impact of either 0.15 for cropland or 0.6 for extensive pasture. In the case of the *Sustainable Development Scenario* in PA, the impact of a conversion on natural habitats, distribution ranges of vertebrate species and, thus calculated BII values could be reduced substantially in comparison to all other scenarios. This effect is attributable to a substantial reduction of the global meat intake which leads to a significant reduction of pasture area which, in turn, can be utilized for an expansion of cropland area. A total of 94% of the cropland expansion was realized on released pasture area (see Table 3), thus decreasing the amount of transformed natural habitat area considerably. Alkimim et al. come to a similar conclusion in their study [87]. In PA, strong decreases of the BII can be witnessed.

In the case of the *Illegal Intensification Scenario* in MT strongly decreasing BII value for threatened vertebrate species and especially small-ranged amphibian species can be observed due to the opening up of protected areas for agricultural expansion. Especially the Pantanal, known for its species richness in regard to birds and amphibians [90], is affected by the displacement of managed land into formerly protected areas. Moreover, we see decreasing BII values for all taxa and categories except endemic vertebrate species which either remain constant or even increase slightly. This can be attributed to the fact that some natural habitat areas that are domiciling endemic vertebrate species are found rather in natural habitats that are not within the protected areas [5]. Since large portions of natural habitats within protected areas are favored for conversion (higher possible crop yield), natural habitats outside these protected areas can be spared from conversion to cultivated land or are converted from cropland to pasture cells due to cropland being relocated to areas within protected habitats (see Table 3). Moreover, the overall higher BII values in the *Legal Intensification Scenario*, compared to the *Illegal Intensification Scenario*, shows that effective conservation of existing protected areas can further enhance biodiversity in MT in 2030 [91]. This observation is confirmed by Sparovek et al. who found the legal command and control frameworks were the most important determinants of conservation outcomes, protecting at least 80% of the existing natural vegetation, and thus biodiversity, in all of their simulated future land use scenarios [85]. In the case of the *Legal Intensification Scenario* in MT, current protected areas are assumed being effectively

conserved, which displaces LULCC from the Pantanal to other not conserved, less biodiverse areas. This prevents strongly decreasing BII value for threatened vertebrate species and especially small-ranged amphibian species in the *Legal Intensification scenario* in contrast to the situation in the *Illegal Intensification Scenario*. As has been discussed before, the expansion of cropland due to a anthropogenic consumption shift towards plant based consumption can only be marginally realized on released pasture area in the *Sustainable Development Scenario* in MT, resulting in BII decreases that are stronger compared to the situation in the *Sustainable Development Scenario* in PA.

Concerning both Federal States, it has to be mentioned that the impact of a conversion of natural habiats on the distribution ranges of vertebrate species expressed by the BII seems relatively high concerning small-ranged and threatened amphibians in PA and small-ranged amphibians in MT. This can be explained by the small extent of the distribution ranges of these vertebrate species [5] (see section 2.4) compared to the other investigated vertebrate species. Only a marginal conversion of natural habitats can mean a severe impact, especially if expressed with the help of an indicator as the BII.

## 3.4. Limitations and uncertainties of the study

First and foremost, a model-based scenario analysis with a focus on environmental impacts should not be misunderstood as a method to predict concrete future events. Rather, it provides a powerful tool to systematically explore plausible constellations of social and economic drivers and the emerging trajectories and dynamics of LULCC, together with its related environmental consequences. It represents a potent method to explore the efficiency and unexpected consequences of policies and can therefore be especially applied to inform decision making under Deep Uncertainty [92]. LandSHIFT has been tested concerning associated uncertainty in modeling LULCC in the Amazon region [58].

Concerning the data that was used to assess the effect of a loss of natural habitat area on species diversity as well as the BII values, the species diversity maps [5], issues of data deficiency have an impact on our estimates. Especially amphibian and mammal species are understudied. Data deficient mammals are mainly concentrated in the Amazon whereas around 30% of all assessed amphibians are generally data deficient [5]. This may lead to an underestimation of the impact of loss of natural habitats on vertebrate diversity especially in regions covered by rainforest (Amazon). Notable are threatened amphibian species. Here, only 4% of the assessed species appear to be threatened, whereas the global rate of threatened amphibians lies at 31% [93] suggesting that the high data deficiency in regard to this taxon and the investigated area are significantly influencing our results.

Moreover, we do not holistically explore the effects of agricultural intensification on natural habitats and its biodiversity. In order to do so it would require an analysis of all factors of agricultural intensification that positively or negatively influences wildlife and habitats. This analysis would have to include emissions caused due to intensification (livestock, fertilizers etc.) and their effect on biodiversity as well as indirect LULCC, for instance caused by a cropland expansion due to an increasing demand for fodder in feedstock systems. Additionally, only vertebrate species are assessed, other representative and important groups of organisms (e.g. insects, soil microorganisms) are not part of this research. Furthermore, the authors simply focus on the assessment based on species numbers. Other determining factors, like functional diversity and functional redundancy, have not yet been incorporated into the analysis.

In all of our scenarios we assume that increases of crop yields until 2030 can be achieved by technological improvements and a more intensive agricultural management alone. At the same time studies such as [55, 94] point out that climate change might have negative effects on

crop yields in Amazonia. It is important to note that this situation might occur until the mid or end of this century when changes in temperature and precipitation are projected to become more intense [95] with potentially stronger negative impacts on crop yields e.g. [94, 96].

## 4. Conclusion

Our results show that especially the potential for agricultural intensification in the Amazon may hint at the way of sustaining food production here [16] in addition to conserving natural habitats from conversion into managed land, thus preserving biodiversity. Moreover, it becomes obvious that especially the change of anthropogenic consumption habits can contribute greatly to safeguard species diversity. This might look like a clear statement in favor of segregative agricultural approaches over integrative agricultural approaches. Segregative approaches focus on increasing agricultural production by an intensified management of agricultural production systems, thus conserving natural habitats from conversion into cropland or pastureland. On the contrary, integrative approaches focus on integrating agricultural production and biodiversity conservation for instance in the form of agroforestry.

However, our results also draw a distressing picture of the future in regard to negative impacts of intensification measures [97]. Despite all the positives of an intensification of agricultural production concerning a conservation of natural habitats, the negative impacts of an intensified agriculture cannot be neglected. Especially pesticide, herbicide, and fertilizer application have to increase in order to increase grass- and cropland productivity [98]. The increased application of agro-chemical products will have negative effects on biodiversity [97]. Especially the use of pesticides in tropical regions has strong negative effects on amphibian populations because they are more susceptible to pesticide use as compared to amphibian populations in temperate regions [99]. Therefore, biodiversity on intensified cropland is likely to decrease [100, 101]. Furthermore, the adoption of intensified agricultural production will be more capital-intensive. If the access to capital is a limiting factor, as might be the case for smallholder farmers, these producers might not be able to apply the required techniques and methods. This in turn will imperil their ability to stay competitive in comparison to large land holders who have better access to monetary resources and can make larger investments into the intensification of agricultural production [102]. Concluding, sufficient access to capital is required to introduce agricultural intensification on a broad scale. An increased livestock production in intensified systems (especially feedstock systems) will increase the demand for livestock fodder production which, in turn, will induce an expansion of cropland area and may be a cause of additional deforestation [102].

Ultimately, future scenario studies will have to adapt a more holistic approach to the assessment of LULCC and agricultural management and its consequences for natural habitats and biodiversity. Currently, this is done, on the one hand, by integrating mentioned organism groups, especially soil microorganisms, and by incorporating the aspects of functional diversity and functional redundancy. On the other hand, scenarios need to be, and will be, constructed with the help of approaches that focus on participatory methods in order to capture stakeholder knowledge as well as developments and drivers especially on local to regional scales thus, increasing the representativeness as well as communicability of results and recommendations.

## Author Contributions

**Conceptualization:** Jan Göpel, Rüdiger Schaldach.

**Formal analysis:** Jan Göpel.

**Funding acquisition:** Rüdiger Schaldach.

**Investigation:** Jan Göpel.

**Methodology:** Jan Schüngel, Benjamin Stuch.

**Project administration:** Rüdiger Schaldach.

**Software:** Jan Schüngel.

**Supervision:** Rüdiger Schaldach.

**Validation:** Jan Göpel.

**Visualization:** Jan Göpel.

**Writing – original draft:** Jan Göpel.

**Writing – review & editing:** Jan Göpel, Benjamin Stuch, Rüdiger Schaldach.

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
