## [Decision Letter · Decision Letter 0]

6 Jan 2020

PONE-D-19-31197

Assessing the effects of agricultural intensification on natural habitats and biodiversity in Southern Amazonia

PLOS ONE

Dear Dr Göpel,

Thank you for submitting your manuscript to PLOS ONE. After careful consideration, we feel that it has merit but does not fully meet PLOS ONE’s publication criteria as it currently stands. Therefore, we invite you to submit a revised version of the manuscript that addresses the points raised during the review process.

We recently received reports from two reviewers that judged the manuscript overall positively, though requiring a deep revision to make it be suitable for publication. In particular, both referees raised concerns about the general structure of the text. Specifically, introduction should be better focused and methods section should be substantially corroborated with important information on modelling input data and relevant variables. We encourage the authors to treasure these two thorough reviews to improve the manuscript as requested.

We would appreciate receiving your revised manuscript by Feb 20 2020 11:59PM. To enhance the reproducibility of your results, we recommend that if applicable you deposit your laboratory protocols in protocols.io, where a protocol can be assigned its own identifier (DOI) such that it can be cited independently in the future. For instructions see: http://journals.plos.org/plosone/s/submission-guidelines#loc-laboratory-protocols

We look forward to receiving your revised manuscript.

Kind regards,

Mirko Di Febbraro

Academic Editor

PLOS ONE

Journal Requirements:

Additional Editor Comments (if provided):

Reviewers' comments:

Reviewer's Responses to Questions

**Comments to the Author**

1. Is the manuscript technically sound, and do the data support the conclusions?

Reviewer #1: Partly

Reviewer #2: Partly

2. Has the statistical analysis been performed appropriately and rigorously? 

Reviewer #1: N/A

Reviewer #2: N/A

3. Have the authors made all data underlying the findings in their manuscript fully available?

Reviewer #1: No

Reviewer #2: No

4. Is the manuscript presented in an intelligible fashion and written in standard English?

Reviewer #1: Yes

Reviewer #2: Yes

5. Review Comments to the Author

Reviewer #1: The authors evaluate whether future agricultural intensification may have an impact on habitats and vertebrate diversity in Mato Grosso and Parà, Brazil. In particular, the authors adopt the LandSHIFT model to simulate the effects of land use and cover changes in combination with additional drivers (e.g. population trend) on natural habitats for threatened, endemic and small-ranged species, and on biodiversity intactness (Biodiversity Intactness Index; BII) in the period 2010-2030 in the two Brazilian regions. The methodological approach and data sources partially build on a previous work by the same authors in the same study area (Göpel et al. 2018; https://doi.org/10.1007/s10113-017-1235-0). Despite the main findings are important to disentangle possible human-driven effects on fragile ecosystems and related biodiversity, the authors need to clarify (or deeper explain) some key aspects at the basis of their work (input data and assumptions), and in turn enhance the overall scientific robustness, innovation and policy relevance of the manuscript. Hereafter some comments and suggestions for improvement from my perspective.

- Formatting pattern. In general, the manuscript is not well balanced among its text parts. I suggest to summarize the last part of the introduction section – this is not a suitable space to describe the methodology (lines 83-101), and better introduce the concepts and current background (lacks and limitations of available studies) of the “effect of a conversion of natural habitats on distribution ranges” and “BII”, possibly in the same region. This can be done by integrating (and enlarging) the text at lines 68-77. I suggest to add a map of the case study area along with section 2.1. The modelling and assessment protocol (section 2.4) should be inserted before the presentation of scenarios and other assumptions, maps and other sources of information then used as inputs for the modelling exercise. This may help the reader to understand the workflow since the beginning. In my opinion, the entire results section (sections 3.1, 3.2, and 3.3) is not very concise and mostly reports what is already summarized in Tables 3 and 4. This can make the reader lost throughout the text, and create misinterpretations when looking at the related discussion. This can be solved by either merging the results and discussion sections, or condensing the results section, for example through deleting any unnecessary element of discussion, which is treated later in the text, thus avoiding repetition. The conclusions section is missing at all. I suggest authors to convey the main findings and possible social and policy implications of the work in this section.

- Land use vs. land cover. In the manuscript, the term “Land Use and Land Cover Change” (LULCC) is widespread (better to specify the full name at the beginning of the introduction section, too). However, one major concern from my side is that authors made a strong assumption on the possible use of land cover instead of land use information in the analysis, which in turn may weaken the discussion related to land use-associated impacts. Indeed, it is not completely clear and transparent to me which data and information sources were used as inputs to the LandSHIFT model (explanatory variables), as well as those related to the dependent variables, such as natural ecosystems and BII. Despite references to model functioning are provided (e.g. Schaldach et al. 2011; https://doi:10.1016/j.envsoft.2011.02.013), in this case, such information (now sparsely provided in the text, see for example lines 179-181, 206, 232-233, 240-249, and Tables 1 and 2) need to be further expanded through e.g., listing all relevant input data, including details on spatial and temporal resolution, and other relevant sources used in the analysis (e.g. population trend), correlated with land use classes (aggregated in Table 2), directly in the text or as supplement. About the dependent variables, it is not clear to me how species ranges are correlated to natural ecosystems, and how natural ecosystems are connected to land use classes used by the model. A table reporting the investigated species (separated in threatened, small-ranged and native) and the correlated natural ecosystems may be useful to deeper understand the effects of a changing in habitats conditions on vertebrates (as reported in the section 3.2). In addition, a clear explanation of how the natural ecosystems were referred to land use classes in the simulation is expected to be provided through e.g., enlarging the text at line 206.

- Scenario assumptions and simulation results. Scenarios are shortly described in section 2.2. Nevertheless, in my opinion, there is a need to further detail which are the variables and parameters (presumably, crop production, forest management intensity, population growth, infrastructures’ density, land tenure, tax system, investments, dietary requirements, etc.) for the explored scenarios – hence affecting the land use – as handled in the modeling exercise, in order to support several sentences in the discussion section which seem now rather speculative (e.g. lines 426-428, 431-433, 448-450, 468-471, 476-478, 490-492, 506-513), and finally to make the methodology transparent enough and replicable. This can be performed by providing a list of variables and parameters, as well as their variations among scenarios’ assumptions, directly in the text or as supplement. This list should be complemented by a critical self-assessment in the the discussion of how the driving forces strictly connected to the scenarios might have led to specific results (e.g. change in dietary needs towards cropland expansion and subsequently habitat loss). Since results are deeply discussed, and the methodology was expected to be rigorous and replicable, it is not sufficient to write in the end that “the inclusion of these factors was beyond the scope of this study” (see also the text before, at lines 533-538). Therefore, a provocative question comes to my mind: did the authors evaluate land use or land cover change impacts on biodiversity? In which way (and robustness) the species abundance is a good explanatory variable for LULCC impacts on biodiversity (see lines 94-97)? Are “agricultural intensification/extensification/compliance with environmental law/changing consumption patterns” (forming the main message of the manuscript and constituting one of the main research questions; see e.g. lines 96-97) referred to land cover elements or land use practices? The reader would expect to see pertinent answers to these questions in the manuscript.

- Communication issues. Results depicted in Figures 1a-3 could be presented in a more readable way (for example, figure captions are missing). I would suggest to modify the graphs as follow: (1) better to transform Figure 1a and 1b into land use change matrices (from 2010 to 2030) for each scenario explored, and put the results for Parà and Mato Grosso in the same figure. This may help the reader to understand the gain and losses (from-to land use classes) and make the differences among scenarios and regions at the same time more explicit. (2) Figures 2 and 3 presumably report absolute numbers of change (between 2010 and 2030) for each species class (threatened, small-ranged, and endemics). Probably, it is more reasonable to report the weighted change (%) over the total (by bar) to understand the individual impact of LULCC on individual species group depending on the scenario. Cross-references to Figures and Tables (as well as to Supplementary Information, if any) need to be established in the text.

- Minor comments. English is fine. Minor typos are spotted throughout the whole text (e.g. lines 197-198) and need to be corrected.

Based on the above-reported comments, authors are asked to carry out major revisions to improve the scientific robustness and clarity of the manuscript, before being completely accepted for publication. Thank you

Reviewer #2: GENERAL COMMENTS

This manuscript regards the use of different land use and cover change scenarios to understand the possible effects on biodiversity (i.e. vertebrates) in two states of Brazil. The topic is of extreme interest because these investigations have great potential in forecasting possible negative outcome of different policies.

The introduction should better reflect your focus and the approach you have adopted. I suggest reporting in the introduction the importance of focusing on vertebrates. For example, vertebrates known contribution to total biodiversity in the Amazon or Brazil or their relevance as umbrella for other groups. Furthermore, there is no explanation on why it would be better to use an indicator representing a trend (however the indicator does not refer to a trend, but it is its change over time that represents a trend) rather than a diversity measure. And the link between the use of a measure of change and biodiversity intactness index is needed. As your work strongly relies on this indicator, I suggest having a paragraph describing it together with its pros and cons.

In the method section it is not clear whether a single land use cover is assigned at the grid-cell level, as for species diversity and ecosystem, when calculating the biodiversity intactness index. I suggest reporting first the methods used to produce the different land use and land cover changes and then the approach to calculate the biodiversity intactness index. This will represent the workflow and the order of your two research questions and will be consistent with the representation of the results. Furthermore, a map of the area investigated is missing.

It is unclear why all figures and tables are reported in the first part of the result section (general one) and not in the specific ones (e.g., table 3 and 4 in the BII subchapter)! I also suggest reducing the text in the result sections; this will enable to underline only the major results that would enable to explain the impact on biodiversity. Indeed, figures are difficult to understand: diagrams overlap with text (in particular figure 2 and 3). Text in the figures is not needed if you then report values in the main text. I must stress that these results are already reported in Fig1 and 2a and 2b of Göpel et al. 2018. Regional Environmental Change 18:129-142 and, in the current state (no citation in the caption of the figure 1), may represent plagiarism. I suggest to represent these results (those of figure 1) in the method section and to focus on the effects on biodiversity in your results – that should be the novelty and focus of the paper. Another option would be to represent data of figure 1 as transition matrices that can give information on total change but also specific information on trends between land covers/uses (but citing in the caption Göpel et al. 2018).

I’m curious to understand why no map was reported representing the BII to understand the distribution of the possible effects of the scenarios. This may enable to give more detailed policy indications to tackle possible scenarios’ effects.

Results reported in table 3 and 4 are not consistent with what reported in the text for the results sections. For example, there are some positive changes for the ILI in Mato Grosso (e.g. endemic birds and mammals in table 4) that are instead not reported as such in the text: “Concerning the Illegal Intensification Scenario, we see decreasing BII values for all taxa and categories” (L404-405). A thorough revision of results is needed as this has influenced the discussion that does cannot be appropriately evaluated based on these inconsistencies. Looking to table 3 and 4 it can be generally stated that SD is the best scenario for biodiversity (especially in Pará) and that ILI has, in general, a lower negative effect than IL. Especially this latter case is strange and deserves additional attention. Finally, language revision is needed.

SPECIFIC COMMENTS

L22: A comma is needed: “..commodities, is required”.

L39-42: The citation should be reported in the text: “ As Martinelli et al. (3) argue”. There are other similar cases in the main text.

L87-88: I suggest reporting and justifying this in the method section. The given justification doesn’t seem to fit as you are focusing on three specific groups that are represented also by wide-range species as well as by endemic and small-ranged species. You analyse these groups, but it becomes clear only when reading the method section!

L192-194: Is this information taken from the global or a national list?

Table 1: These values are the same for all groups of species? Is this the case also for the given citations?

L233-236: It is not clear if the citations refer to works that made such simplifications or are simply citations of the more detailed classifications?

L375-413: I suggest to produce maps or to avoid reporting in the text data presented in the mentioned tables (by the way these are not consistent, see general comments).

L482-484: “The positive implication of agricultural intensification on biodiversity found in PA is confirmed also in MT. Here, the BII values decrease in almost all assessed taxa and categories in the case of the Legal Intensification Scenario and Illegal Intensification Scenario.” These two sentences seem to be contrasting each other. How can a decrease in BII be described as positive implication on biodiversity? See also general comments.

6. PLOS authors have the option to publish the peer review history of their article (what does this mean?). If published, this will include your full peer review and any attached files.

Reviewer #1: No

Reviewer #2: No

---

## [Author Response · Author response to Decision Letter 0]

11 Apr 2020

Response to Reviewer #1:

Thank you very much for the time you invested into reviewing our manuscript. We tried to consider all of your suggestions. If this consideration was not possible, we explain the reasons. 

7. Formatting pattern. In general, the manuscript is not well balanced among its text parts. I suggest to summarize the last part of the introduction section – this is not a suitable space to describe the methodology (lines 83-101), and better introduce the concepts and current background (lacks and limitations of available studies) of the “effect of a conversion of natural habitats on distribution ranges” and “BII”, possibly in the same region. This can be done by integrating (and enlarging) the text at lines 68-77. 

- We reworked the introduction section considerably, especially deleting any information that might resemble information concerning methods with one exception. Reviewer 2 requested us to include a paragraph describing the BII. We did so but kept this paragraph intentionally short in order to comply with both reviewers’ requests (lines 97 – 106). In addition, we streamlined the introduction section in addition to including more information on other studies in the region as well as lacks and shortcomings of the mentioned studies. (lines 57 -96)

8. I suggest to add a map of the case study area along with section 2.1. 

- Done. (line 113, Fig. 1)

9. The modelling and assessment protocol (section 2.4) should be inserted before the presentation of scenarios and other assumptions, maps and other sources of information then used as inputs for the modelling exercise. This may help the reader to understand the workflow since the beginning. 

- Done. (section 2.2)

10. In my opinion, the entire results section (sections 3.1, 3.2, and 3.3) is not very concise and mostly reports what is already summarized in Tables 3 and 4. This can make the reader lost throughout the text, and create misinterpretations when looking at the related discussion. This can be solved by either merging the results and discussion sections, or condensing the results section, for example through deleting any unnecessary element of discussion, which is treated later in the text, thus avoiding repetition. The conclusions section is missing at all. I suggest authors to convey the main findings and possible social and policy implications of the work in this section.

- The results section was completely reworked in addition to re-doing modeling runs and the required analysis (sections 3.1, 3.2, 3.3). We changed the structure of the manuscript, cleaning the specific results sections (text) of any elements of discussion and split up the original discussion section in order to be able to add a specific discussion section to each specific results section. Thereby, we hope to have eliminated any confusing elements in addition to increasing the flow when reading the manuscript in its new version. Furthermore, we added a conclusion section and structured it according to the recommendations (section 4).

11. Land use vs. land cover. In the manuscript, the term “Land Use and Land Cover Change” (LULCC) is widespread (better to specify the full name at the beginning of the introduction section, too).

- Done. (line 25 and line 40)

12. However, one major concern from my side is that authors made a strong assumption on the possible use of land cover instead of land use information in the analysis, which in turn may weaken the discussion related to land use-associated impacts. 

- We made clear that we are referring to “land-use change” throughout the document eliminating any mention of land-cover change where appropriate despite the fact that LandSHIFT is no pure land-use model but also elucidates the change of one land-cover to another. However, Reviewer 1 was correct in his assessment of the confusing use of land-cover change instead of land-use change in the context of this specific analysis. We acted accordingly.

13. Indeed, it is not completely clear and transparent to me which data and information sources were used as inputs to the LandSHIFT model (explanatory variables), as well as those related to the dependent variables, such as natural ecosystems and BII. Despite references to model functioning are provided (e.g. Schaldach et al. 2011; https://doi:10.1016/j.envsoft.2011.02.013), in this case, such information (now sparsely provided in the text, see for example lines 179-181, 206, 232-233, 240-249, and Tables 1 and 2) need to be further expanded through e.g., listing all relevant input data, including details on spatial and temporal resolution, and other relevant sources used in the analysis (e.g. population trend), correlated with land use classes (aggregated in Table 2), directly in the text or as supplement. 

- The data that was requested by Reviewer 1 cannot be provided directly in this manuscript. All the relevant data is already published in numerous studies and falls under the copyright of the respective publishers. Throughout the text we added specific information in regard to input data used to initiate and drive the model (lines 163-164). This information can be accessed easily. The same applies to the socioeconomic assumptions of the scenarios (open access) (line 193). Correlation of land-use classes has been elaborated (lines 167-170).

14. About the dependent variables, it is not clear to me how species ranges are correlated to natural ecosystems, and how natural ecosystems are connected to land use classes used by the model. A table reporting the investigated species (separated in threatened, small-ranged and native) and the correlated natural ecosystems may be useful to deeper understand the effects of a changing in habitats conditions on vertebrates (as reported in the section 3.2). In addition, a clear explanation of how the natural ecosystems were referred to land use classes in the simulation is expected to be provided through e.g., enlarging the text at line 206.

- We tried to explain in more detail how species information and land-use information come together by reworking the appropriate section (lines 173-178). We did not include a table containing the assessed species in detail. Referring to section 2.4 of the manuscript, the sheer number of species integrated in the study made this not possible: 1703 bird species, 637 mammal species and 875 amphibian species were considered in this study. Ecosystems and land-use classes were correlated according to Friedl et al. (2010) (lines 165-170).

15. Scenario assumptions and simulation results. Scenarios are shortly described in section 2.2. Nevertheless, in my opinion, there is a need to further detail which are the variables and parameters (presumably, crop production, forest management intensity, population growth, infrastructures’ density, land tenure, tax system, investments, dietary requirements, etc.) for the explored scenarios – hence affecting the land use – as handled in the modeling exercise, in order to support several sentences in the discussion section which seem now rather speculative (e.g. lines 426-428, 431-433, 448-450, 468-471, 476-478, 490-492, 506-513), and finally to make the methodology transparent enough and replicable. This can be performed by providing a list of variables and parameters, as well as their variations among scenarios’ assumptions, directly in the text or as supplement. This list should be complemented by a critical self-assessment in the the discussion of how the driving forces strictly connected to the scenarios might have led to specific results (e.g. change in dietary needs towards cropland expansion and subsequently habitat loss). 

- An elaborate description and discussion of the translation process from qualitative to quantitative information and the respective results for all scenario assumptions is described in Schönenberg R, Schaldach R, Lakes T, Göpel J, Gollnow F. Inter- and transdisciplinary scenario construction to explore future land-use options in southern Amazonia. E&S 2017; 22(3). Here the translation process of qualitative information is described in addition to the results including changes of the integrated and considered factors over time as well as information on the dimension of agreement with other renowned studies in this field. We could not integrate this information in this manuscript as the information is under copyright, having been published in other journals (lines 193 and 198). Moreover, the discussion sections were re-written to better reflect the influence of socioeconomic driving forces on modeling and assessment results (sections 3.1.3, 3.2.3, 3.3.3).

16. Since results are deeply discussed, and the methodology was expected to be rigorous and replicable, it is not sufficient to write in the end that “the inclusion of these factors was beyond the scope of this study” (see also the text before, at lines 533-538). 

- Specific line has been omitted. Moreover, we included thoughts and plans for further research and studies (lines 566-575; section 4).

17. Therefore, a provocative question comes to my mind: did the authors evaluate land use or land cover change impacts on biodiversity? In which way (and robustness) the species abundance is a good explanatory variable for LULCC impacts on biodiversity (see lines 94-97)? Are “agricultural intensification/extensification/compliance with environmental law/changing consumption patterns” (forming the main message of the manuscript and constituting one of the main research questions; see e.g. lines 96-97) referred to land cover elements or land use practices? The reader would expect to see pertinent answers to these questions in the manuscript.

- A good point. We agree that this has not been made clear throughout the manuscript. We tried to elucidate the information by, on the one hand, including information to what extent species abundance is a good explanatory variable for LULCC impacts on biodiversity (lines 75-87). On the other hand, we tried to clear the impression of referring to land cover elements when focusing on socioeconomic drivers of land-use change (including compliance with the law) (sections 3.1.3, 3.2.3, 3.3.3). Lastly, we included elaborate information concerning land-use change being the driving force behind impacts on biodiversity throughout the manuscript.

18. Communication issues. Results depicted in Figures 1a-3 could be presented in a more readable way (for example, figure captions are missing). I would suggest to modify the graphs as follow: (1) better to transform Figure 1a and 1b into land use change matrices (from 2010 to 2030) for each scenario explored, and put the results for Parà and Mato Grosso in the same figure. This may help the reader to understand the gain and losses (from-to land use classes) and make the differences among scenarios and regions at the same time more explicit.

- We included a land-use change matrix (PA and MT included in 1 table) (Table 4) in order to better build our argumentation on the respective changes from one land-use class to another. We hope that especially the main messages as well as the discussion sections benefit from this.

19. (2) Figures 2 and 3 presumably report absolute numbers of change (between 2010 and 2030) for each species class (threatened, small-ranged, and endemics). Probably, it is more reasonable to report the weighted change (%) over the total (by bar) to understand the individual impact of LULCC on individual species group depending on the scenario. Cross-references to Figures and Tables (as well as to Supplementary Information, if any) need to be established in the text.

- We kept to our method of displaying the total area affected by land-use change but tried to present it in a more readable and reasonable fashion (Figures 2 and 3). Furthermore, we calculated the area-weighted change (%) of the affected habitats known as distribution ranges of vertebrate species and presented these results in the form of text in the respective results & discussion section (sections 3.2.1 and 3.2.2).

20. Minor comments. English is fine. Minor typos are spotted throughout the whole text (e.g. lines 197-198) and need to be corrected.

- We corrected the minor typing mistakes throughout the document. Furthermore, we had an external provider check the language. 

 

Response to Reviewer #2:

We thank you very much for taking the time to thoroughly reviewing the manuscript and we tried to consider your recommendations but explained where this was not possible.

21. The introduction should better reflect your focus and the approach you have adopted. I suggest reporting in the introduction the importance of focusing on vertebrates. For example, vertebrates known contribution to total biodiversity in the Amazon or Brazil or their relevance as umbrella for other groups.

- We integrated further information on why it is appropriate to use vertebrate species, specifically the vertebrate species we assessed, as a proxy for biodiversity loss (lines 75-87). Furthermore, we not only assess the reduction of vertebrate species as a proxy for biodiversity loss, we also refer to the loss of natural habitats and therefore natural vegetation in our research (section 3.2.x). Therefore, we use the decrease of plant- and wildlife as a proxy for biodiversity. Nonetheless, we critically described missing factors and neglected species in section 3.4 of the manuscript.

22. Furthermore, there is no explanation on why it would be better to use an indicator representing a trend (however the indicator does not refer to a trend, but it is its change over time that represents a trend) rather than a diversity measure. And the link between the use of a measure of change and biodiversity intactness index is needed. As your work strongly relies on this indicator, I suggest having a paragraph describing it together with its pros and cons.

- We included the information in the introduction section as requested (lines 97-106). We would have liked to better integrate this information in section 3.4 but we also see that we build the argumentation heavily on especially this indicator and thus, have made sure to inform accordingly in the beginning of the manuscript.

23. In the method section it is not clear whether a single land use cover is assigned at the grid-cell level, as for species diversity and ecosystem, when calculating the biodiversity intactness index. I suggest reporting first the methods used to produce the different land use and land cover changes and then the approach to calculate the biodiversity intactness index. This will represent the workflow and the order of your two research questions and will be consistent with the representation of the results. Furthermore, a map of the area investigated is missing.

- We have restructured the methods section (section 2.2) accordingly and now hope to be able to better inform about what steps we took and how we approached this research.

24. It is unclear why all figures and tables are reported in the first part of the result section (general one) and not in the specific ones (e.g., table 3 and 4 in the BII subchapter)! I also suggest reducing the text in the result sections; this will enable to underline only the major results that would enable to explain the impact on biodiversity. Indeed, figures are difficult to understand: diagrams overlap with text (in particular figure 2 and 3). Text in the figures is not needed if you then report values in the main text. I must stress that these results are already reported in Fig1 and 2a and 2b of Göpel et al. 2018. Regional Environmental Change 18:129-142 and, in the current state (no citation in the caption of the figure 1), may represent plagiarism. I suggest to represent these results (those of figure 1) in the method section and to focus on the effects on biodiversity in your results – that should be the novelty and focus of the paper. Another option would be to represent data of figure 1 as transition matrices that can give information on total change but also specific information on trends between land covers/uses (but citing in the caption Göpel et al. 2018).

- We have deleted any information that might be misinterpreted as plagiarism. Furthermore, we cleaned such elements as diagrams overlapping with text. We completely rewrote most of the manuscript in order to not report in the text what is already discernable in the tables and graphs. A transition matrix (Table 4) is included and we refer to Göpel et al. (2018) throughout the text. We did not cite Göpel et al. (2018) in the caption of Table 4 as the way of presenting the information does not require this step anymore alongside the fact that we modeled from scratch in order to inform this study and manuscript.

25. I’m curious to understand why no map was reported representing the BII to understand the distribution of the possible effects of the scenarios. This may enable to give more detailed policy indications to tackle possible scenarios’ effects.

- True. This requires more work that has already been started according to the comments of Reviewer 2. Currently we are integrating code into the model that calculates the BII according to the land-use change happening for each single grid cell and each calculated time step (every 5 years) and will be a part of the model output in the form of spatially explicit maps. To integrate this method, we need, as mentioned, to rewrite code as well as check its functioning in the interplay of all modules of the land-use change model, which is quite tedious and time consuming and therefore has not been integrated in this study.

26. Results reported in table 3 and 4 are not consistent with what reported in the text for the results sections. For example, there are some positive changes for the ILI in Mato Grosso (e.g. endemic birds and mammals in table 4) that are instead not reported as such in the text: “Concerning the Illegal Intensification Scenario, we see decreasing BII values for all taxa and categories” (L404-405). A thorough revision of results is needed as this has influenced the discussion that does cannot be appropriately evaluated based on these inconsistencies. 

- This study required a comprehensive and thorough revision and we are most grateful for the thorough review by both reviewers. We not only revised the manuscript, we also completely re-did the modeling work by re-running the land-use change model for each scenario and re-analyzing the output. We restructured the whole manuscript, re-wrote especially the results sections and concentrated the discussion sections (now integrated into the results) with a focus on the results that were reported in the respective results sections.

27. Looking to table 3 and 4 it can be generally stated that SD is the best scenario for biodiversity (especially in Pará) and that ILI has, in general, a lower negative effect than IL. Especially this latter case is strange and deserves additional attention.

- In general, SD is truly the scenario in terms of impacts on biodiversity, except for MT, where the missing release of pasture area and the strong increase of the demand for plant based food leads to, especially, BII values that are not very different from both intensification scenarios. After re-analyzing and modeling from scratch, we found that you might be looking at different results now. The positive effect, in terms of a conservation of plant- and wildlife, in the LI Scenario is actually slightly stronger compared to ILI (also see the respective discussion sections).

28. Finally, language revision is needed.

- Done.

29. L22: A comma is needed: “..commodities, is required”.

- Done.

30. L39-42: The citation should be reported in the text: “ As Martinelli et al. (3) argue”. There are other similar cases in the main text.

- Done, throughout the text.

31. L87-88: I suggest reporting and justifying this in the method section. The given justification doesn’t seem to fit as you are focusing on three specific groups that are represented also by wide-range species as well as by endemic and small-ranged species. You analyse these groups, but it becomes clear only when reading the method section!

- Done, now section 2.4.

32. L192-194: Is this information taken from the global or a national list?

- Information was taken from the national list. Information and justification see (lines 250-251).

33. Table 1: These values are the same for all groups of species? Is this the case also for the given citations?

- Yes, these values are the same for all investigated species, also in the mentioned publications.

34. L233-236: It is not clear if the citations refer to works that made such simplifications or are simply citations of the more detailed classifications?

- These are just citations of the more detailed classifications. The authors aggregated the detailed classes into less detailed classes in order to focus not on specific forest types but on ecosystems as a whole. 

35. L375-413: I suggest to produce maps or to avoid reporting in the text data presented in the mentioned tables (by the way these are not consistent, see general comments).

- As mentioned before, the results sections were re-written in order to not state in the text was is discernable in the respective tables and figures.

36. L482-484: “The positive implication of agricultural intensification on biodiversity found in PA is confirmed also in MT. Here, the BII values decrease in almost all assessed taxa and categories in the case of the Legal Intensification Scenario and Illegal Intensification Scenario.” These two sentences seem to be contrasting each other. How can a decrease in BII be described as positive implication on biodiversity? See also general comments.

- Absolutely true and corrected.

---

## [Decision Letter · Decision Letter 1]

12 May 2020

PONE-D-19-31197R1

Assessing the effects of agricultural intensification on natural habitats and biodiversity in Southern Amazonia

PLOS ONE

Dear Dr Göpel,

Thank you for submitting your manuscript to PLOS ONE. After careful consideration, we feel that it has merit but does not fully meet PLOS ONE’s publication criteria as it currently stands. Therefore, we invite you to submit a revised version of the manuscript that addresses the points raised during the review process.

Both referees greatly appreciated your effort in amending the manuscript according to their suggestion. Some minor points still remain to be fixed (see the report below), though I am pretty sure you will not have any problem in addressing them.

We would appreciate receiving your revised manuscript by Jun 26 2020 11:59PM. To enhance the reproducibility of your results, we recommend that if applicable you deposit your laboratory protocols in protocols.io, where a protocol can be assigned its own identifier (DOI) such that it can be cited independently in the future. For instructions see: http://journals.plos.org/plosone/s/submission-guidelines#loc-laboratory-protocols

We look forward to receiving your revised manuscript.

Kind regards,

Mirko Di Febbraro

Academic Editor

PLOS ONE

Reviewers' comments:

Reviewer's Responses to Questions

**Comments to the Author**

1. If the authors have adequately addressed your comments raised in a previous round of review and you feel that this manuscript is now acceptable for publication, you may indicate that here to bypass the “Comments to the Author” section, enter your conflict of interest statement in the “Confidential to Editor” section, and submit your "Accept" recommendation.

Reviewer #1: All comments have been addressed

Reviewer #2: (No Response)

2. Is the manuscript technically sound, and do the data support the conclusions?

Reviewer #1: Yes

Reviewer #2: Yes

3. Has the statistical analysis been performed appropriately and rigorously? 

Reviewer #1: Yes

Reviewer #2: Yes

4. Have the authors made all data underlying the findings in their manuscript fully available?

Reviewer #1: Yes

Reviewer #2: No

5. Is the manuscript presented in an intelligible fashion and written in standard English?

Reviewer #1: Yes

Reviewer #2: Yes

6. Review Comments to the Author

Reviewer #1: I thank the authors for their efforts to address all issues raised. Very appreciated! I do not have additional comments.

Authors could consider to move text from lines 97-106 before line 88. Authors could check some typos throughout the text, such e.g. at lines 260 ("vertebrate").

Reviewer #2: I thank the authors for considering all my comments. Authors made some important changes based on suggestions of both reviewers and I think the manuscript has greatly improved compared to the former version.

However, I still have some minor comments.

i. The inclusion of small “discussion” subsections in the results chapter is a bit strange but I understand this was done to follow comments by Rev1. However, there is no such subsection for chapter 3.1. I suggest changing the name of the small “discussion” subsections with a more informative title (and, hence, avoiding having two subsections with the same name).

ii. The discussion parts require some additional reference to words dealing with future scenarios. Comparisons with similar studies for Amazonia area are needed.

iii. I also think that yours is a “strong” main conclusion at least when reading the abstract! Caution on intensification is, however, highlighted in the conclusion section (I would also stress that results can change between different regions as the two case studies highlighted). These results point out to preferring segregative approaches rather than integrative approaches. I think this (land sparing vs. land sharing and segregative vs. integrative) should be mentioned in the discussion or conclusion section as this work contributes to this “hot topic”.

iv. There are a number of typos and mistakes throughout the text. As examples, L535:..”areas the are domicile to”? L558: See section 1? Did authors mean 3.1? L531: “(86)(5),” in “(5, 86)”?

v. L455-460: I suggest dividing this long sentence.

vi. Why “Deep-Uncertainty” with capital letters? Citation needed?

vii. Several typos in the Reference list (e.g. L632, L656, L660,..: lack of capital letters in journals’ names.

7. PLOS authors have the option to publish the peer review history of their article (what does this mean?). If published, this will include your full peer review and any attached files.

Reviewer #1: No

Reviewer #2: No

---

## [Author Response · Author response to Decision Letter 1]

1 Jul 2020

• First off, we have to thank the reviewers again for taking the time and risking the nerve while reviewing our manuscript. Your input has considerably helped to improve the manuscript as a whole and the statements made therein. 

6. Review Comments to the Author

Reviewer #1: I thank the authors for their efforts to address all issues raised. Very appreciated! I do not have additional comments.

• We appreciate the time both reviewers took to thoroughly go through the paper and the extended advice you both gave to make this paper more profound. Responding in kind was the only possible response.

Authors could consider to move text from lines 97-106 before line 88. Authors could check some typos throughout the text, such e.g. at lines 260 ("vertebrate").

• We have moved the corresponding the explanatory text concerning to a more appropriate spot in the introduction (former l. 97-106, now l. 88-97)

• We have corrected several typos (i.e. “vertebarte” to vertebrate, l. 270)

Reviewer #2: I thank the authors for considering all my comments. Authors made some important changes based on suggestions of both reviewers and I think the manuscript has greatly improved compared to the former version.

However, I still have some minor comments.

i. The inclusion of small “discussion” subsections in the results chapter is a bit strange but I understand this was done to follow comments by Rev1. However, there is no such subsection for chapter 3.1. I suggest changing the name of the small “discussion” subsections with a more informative title (and, hence, avoiding having two subsections with the same name).

• We have changed the titles of the discussion subsections to something more explanatory and hope to increase the informational content now given by the title. (l. 440-442; 522-524)

• We did not want to add another subsection in chapter 3.1 because the discussion regarding the content of the respective chapter has been performed thoroughly in another publication. Further, chapter 3.1 is just (as both reviewers have acknowledged correctly) a “summary” of mentioned other publication and is placed in this publication to again point out the background for the subsequent analysis.

ii. The discussion parts require some additional reference to words dealing with future scenarios. Comparisons with similar studies for Amazonia area are needed.

• We have included several further scenario studies for the Amazon region in both discussion subsections. (l. 448-452; 455-459; 472-474; 550-551; 566-569)

• Further, some studies (i.e. Galford et al.) used as a “benchmark” in this paper have been conducted as scenario studies but have not been mentioned specifically as “future scenario studies” within this manuscript.

iii. I also think that yours is a “strong” main conclusion at least when reading the abstract! Caution on intensification is, however, highlighted in the conclusion section (I would also stress that results can change between different regions as the two case studies highlighted). These results point out to preferring segregative approaches rather than integrative approaches. I think this (land sparing vs. land sharing and segregative vs. integrative) should be mentioned in the discussion or conclusion section as this work contributes to this “hot topic”.

• Reviewer 2 mentions something we have been discussing in length. We did not want to put the focus on that specific discussion due to several reasons. Although I specifically felt this study to contribute extensively to said discussion. So, I´m happy with having actually been asked to say at least something in contribution to the “land sparing vs. land sharing” debate. (l. 627-632)

iv. There are a number of typos and mistakes throughout the text. As examples, L535:..”areas the are domicile to”? L558: See section 1? Did authors mean 3.1? L531: “(86)(5),” in “(5, 86)”?

• The mentioned typos have been corrected. One should expect something like that occurring being at least unlikely after a thorough and professional spelling check. Thanks a lot.

• We deleted the reference to section 1 in l. 558. We actually wanted to refer to that section as it incorporates some negative aspects of employing the BII. After consideration we decided that reference to section 1 isn´t fitting here. 

v. L455-460: I suggest dividing this long sentence.

• We have divided the mentioned sentence. (l. 478-482)

vi. Why “Deep-Uncertainty” with capital letters? Citation needed?

• Yes, a citation was needed here as we refer to a specific concept framed in a specific publication. Thank you for the hint. (l. 539)

vii. Several typos in the Reference list (e.g. L632, L656, L660,..: lack of capital letters in journals’ names.

The lack of capital letters (journal names) in the reference list have been resolved.

---

## [Editor Report · Decision Letter 2]

6 Jul 2020

Assessing the effects of agricultural intensification on natural habitats and biodiversity in Southern Amazonia

PONE-D-19-31197R2

Dear Dr. Göpel,

We’re pleased to inform you that your manuscript has been judged scientifically suitable for publication and will be formally accepted for publication once it meets all outstanding technical requirements.

Kind regards,

Mirko Di Febbraro

Academic Editor

PLOS ONE